# Silvicultural Interventions Drive the Changes in Soil Organic Carbon in Romanian Forests According to Two Model Simulations

**Viorel N. B. Blujdea** [1,*] , **Toni Viskari** [2], **Liisa Kulmala** [2,3] , **George Gârbacea** [4], **Ioan Dutcă** [1,5] , **Mihaela Miclăuș** [1], **Gheorghe Marin** [1,4] **and Jari Liski** [2]

1   Department of Silviculture, Transilvania University of Brasov, Sirul Beethoven, 1, 500123 Brasov, Romania; idutca@unitbv.ro (I.D.); miclaus.miha@gmail.com (M.M.); gh_marin_icas@yahoo.com (G.M.)
2   Finnish Meteorological Institute, Erik Palménin Aukio 1, FI-00560 Helsinki, Finland; Toni.Viskari@fmi.fi (T.V.); liisa.kulmala@fmi.fi (L.K.); Jari.Liski@fmi.fi (J.L.)
3   Institute for Atmospheric and Earth System Research (INAR)/Forest Sciences, University of Helsinki, FI-00014 Helsinki, Finland
4   National Institute for Research and Development in Silviculture "Marin Dracea", Eroilor 128, 077190 Voluntari, Romania; george.garbacea@yahoo.com
5   Buckinghamshire New University, Queen Alexandra Rd, High Wycombe HP11 2JZ, UK
*   Correspondence: viorel.blujdea@unitbv.ro

**Abstract:** We investigated the effects of forest management on the carbon (C) dynamics in Romanian forest soils, using two model simulations: CBM-CFS3 and Yasso15. Default parametrization of the models and harmonized litterfall simulated by CBM provided satisfactory results when compared to observed data from National Forest Inventory (NFI). We explored a stratification approach to investigate the improvement of soil C prediction. For stratification on forest types only, the NRMSE (i.e., normalized RMSE of simulated vs. NFI) was approximately 26%, for both models; the NRMSE values reduced to 13% when stratification was done based on climate only. Assuming the continuation of the current forest management practices for a period of 50 years, both models simulated a very small C sink during simulation period (0.05 MgC ha$^{-1}$ yr$^{-1}$). Yet, a change towards extensive forest management practices would yield a constant, minor accumulation of soil C, while more intensive practices would yield a constant, minor loss of soil C. For the maximum wood supply scenario (entire volume increment is removed by silvicultural interventions during the simulated period) Yasso15 resulted in larger emissions ($-0.3$ MgC ha$^{-1}$ yr$^{-1}$) than CBM ($-0.1$ MgC ha$^{-1}$ yr$^{-1}$). Under 'no interventions' scenario, both models simulated a stable accumulation of C which was, nevertheless, larger in Yasso15 (0.35 MgC ha$^{-1}$ yr$^{-1}$) compared to CBM-CSF (0.18 MgC ha$^{-1}$ yr$^{-1}$). The simulation of C stock change showed a strong "start-up" effect during the first decade of the simulation, for both models, explained by the difference in litterfall applied to each scenario compared to the spinoff scenario. Stratification at regional scale based on climate and forest types, represented a reasonable spatial stratification, that improved the prediction of soil C stock and stock change.

**Keywords:** model intercomparison; CBM-CFS3; Yasso15; silvicultural scenarios; litterfall dynamic





## 1. Introduction

Soil is the common element across ecosystems, from the natural to intensely anthropogenically modified ones. Due to societal needs, soil is modified through a range of disturbances, from direct and strong (e.g., for crops or infrastructure constructions) to indirect and light (e.g., through intervention on vegetation like in extensive grazing). Soil represents the largest biogeochemically active terrestrial carbon pool on Earth [1,2] by storing some 2300 Pg of carbon (C) down to 3 m soil depth [3]. Globally, soils are a $CO_2$ sink [4] but locally, both natural and human-induced disturbances affect the carbon balance in both ways.

In Europe, the soil C stocks appear to only change slightly even for the most exposed land categories, like agricultural lands under a range of climate change scenarios [5,6]. Within forests, the soils show a contribution to atmospheric exchange proportional to forest area [7,8], generally comparable to grasslands [9].

Sudden changes affecting forests, i.e., deforestation or natural disturbances, impact the soil C stocks and greenhouse gas (GHG) emissions, sometime with significant magnitudes. Historically, the cumulated loss of C from deforestation was an important driver of increasing $CO_2$ concentration in the atmosphere [10]. By opposition, slow changes affect C stocks and GHG emissions from all forest pools, either negatively through insidious degradation [11] or positively through gains by afforestation and restoration of degraded forests [12]. In sustainably managed forest ecosystems, the soil carbon pool is often overlooked [13] given its low contribution to forest $CO_2$ sink, despite that it generally represents a higher share of total carbon stock of the forests and for most of the other terrestrial ecosystems [1].

Lately, growing interest on the quantification of the GHG fluxes between the soil pool and atmosphere was driven by requirement to report anthropogenic GHG emissions and $CO_2$ removals as part of the national GHG inventory under United Nations Framework Convention on Climate Change (UNFCCC) process for all land uses [14–16]. Given the rather stable pattern of forestry interventions at the national scale, there is a generally reasonable assumption that the C stocks remain rather constant in time, i.e., Tier 1 assumption under Intergovernmental Panel on Climate Change (IPCC) [14,17]. Management changes and evolving climate change nevertheless challenges this approach [18–20]. Moreover, the participation in the emission reduction policy requires understanding and quantifying carbon stocks changes and non-$CO_2$ fluxes from soils, i.e., the impact of forest management practices and natural disturbances. The practical implementation of GHG mitigation mechanisms requires subnational scale of the estimation, e.g., regional, local or ownership scale. A significant push for consistent soil carbon data was driven by including forest management on the list of eligible activities for emission reduction under Marrakesh Accords [21]. Since then, soil was maintained throughout all instruments, e.g., in the two commitment periods of the Kyoto Protocol 2008–2012 and 2013–2020 [22], and finally under Paris Agreement [23], e.g., through Regulation (EU)2018/841, so called Land use, land use change and forestry (LULUCF) Regulation, applicable to the member states of the European Union.

The calculation of the C stocks requires multiple empirical parameters like C content, soil apparent density and rock content for the relevant depth, while all of them are affected by uncertainty given the sampling scheme and processing method [24,25] or the forests' particularity and spatial fragmentation [26]. Although some countries have long time series of robust monitoring and data on C stocks in all carbon pools of forests [27], they often have limited information on short term C stock change in mineral soils. As a result, they face challenges in using the available datasets to their full potential, e.g., for topsoil organic carbon content across Europe [28]. The few existing repeated national forests soil monitoring systems report a wide range of short time changes in soil C: loss in England and Wales [29] and gain in Finland [30] and France [19]. The large uncertainty of the estimates, though, makes the short time change almost undetectable. For example, according to Danish inventory design [31], the annual C stock changes must exceed 0.15 MgC ha$^{-1}$ y$^{-1}$ to be detected.

Models are often mentioned as suitable and economically convenient solutions to ensure soil related GHG reporting [17,32,33]. Most forest soil carbon models are driven by national forest inventories (NFI) data and need soil measurement from at least one moment in time for calibration and validation.

The two models we used in this paper are CBM-CFS3 and Yasso15. They are used for both advancing the understanding of soil processes [32], for GHG inventory reporting, i.e., Yasso15 in Austria and Finland and CBM in Ireland, or CBM for analysis of mitigation pathways in EU [33], or Canada [34]. Both are tools for projecting C stocks in forest mineral

soils, while CBM allows enhanced representation of all key ecological processes, e.g., biomass growth and soils decomposition [35]. Yasso15 performed satisfactorily in various inter-model comparisons (for Finland [30,36]), calibration by litter bag decomposition experiments [17] or against measured data [37]. CBM-CFS3 provides a resolution at the level of 11 dead organic matter pools which allows matching to the three pools defined by [14], namely dead wood, litter and soils organic matter.

Romanian forests have a strong altitudinal distribution, which is reflected in the climate, vegetation and soil properties [38]. Most of the existing studies in Romania are focused on soils' spatial and geographical distribution [39] and few on the C stocks [40,41]. However, robust data on short term C stock changes is still missing.

Romania completed the first systematic forest soil inventory as part of the national GHG inventory effort for LULUCF sector. The soil sampling scheme is fully embedded and run as part of the National Forest Inventory (NFI) framework [42–44].

The aim of this study was to understand the soil organic carbon dynamics in Romanian forests under the impact of various forest management practices. Within this study we addressed three specific research questions:

(1) Does including detailed soil organic carbon dynamic models, i.e., running carbon pools by CBM and chemical compounds by Yasso15, improve the simulations of the initialized C stock compared to measured ones?
(2) Do models perform comparatively on short term assuming the same litterfall dynamic?
(3) How do different harvesting scenarios for Romania's forest affect the carbon simulations?

For these purposes, we assessed the sensitivity of the CBM and Yasso15 models to harmonized biomass inputs and temperature at regional/local scale and compared simulated to measured NFI data. From a scientific perspective, we focused on how including a more detailed SOC estimation in the model initialization improves the model performance at regional vs. national scale, assuming default models parametrization.

## 2. Materials and Methods

### 2.1. Description of Soil Modules of CBM-CFS3 and Yasso15

Both models run with annual time step and use litterfall and climate as driver data, while do not require other information on soil physical and chemical properties. CBM runs C pools, while Yasso15 runs biochemical compounds (Figure 1), while both are limited to simulation of mineral soils only.

CBM-CFSv3 (CBM) is a forest carbon model for spatial, stand- and landscape-level dynamics [35]. CBM implements forest growth based on volume increment and conversion of volume to biomass, while estimates the litter inputs based on turnovers for each living biomass compartment. In old or unmanaged stands, the loss of living biomass due to natural processes represents additional mortality in the model. It incorporates a soil model which tracks nine dead organic matter subpools which strive to describe the decomposition process relative to (i) type of biomass input with annual time step (which refers to dead organic matter particles dimensions), (ii) forest species grouping (only for standing dead wood, i.e., snags in hardwood and softwood), (iii) positioning of decomposition above or belowground soil surface, and (iv) relative decay rate for each subpool according to four degrees (very fast, fast, medium and slow). According to Kurz et al. [35], the decomposition is modeled for each subpool by applying two factors to the base decay rate for the reference mean annual average temperature of 10 °C: (i) temperature-dependent decay modifier (which usually reduces the decomposition rate) and (ii) an open-canopy effect decay multiplier reflecting the stand characteristics (which usually enhances the decomposition rate). As effect, approximately 83% of the C lost by any subpool is converted to $CO_2$ emitted to atmosphere in time step of one year, while the rest is stored or transferred to other subpools. Physical transfers among certain subpools apply to each time step, e.g., from coarse to intermediary medium or fast, or from aboveground to belowground subpools. Specifically, CBM version we used to allow only one unique set of decomposition

factors for all forest types and climates. How climate influences the decomposition is described for CBM by [35].

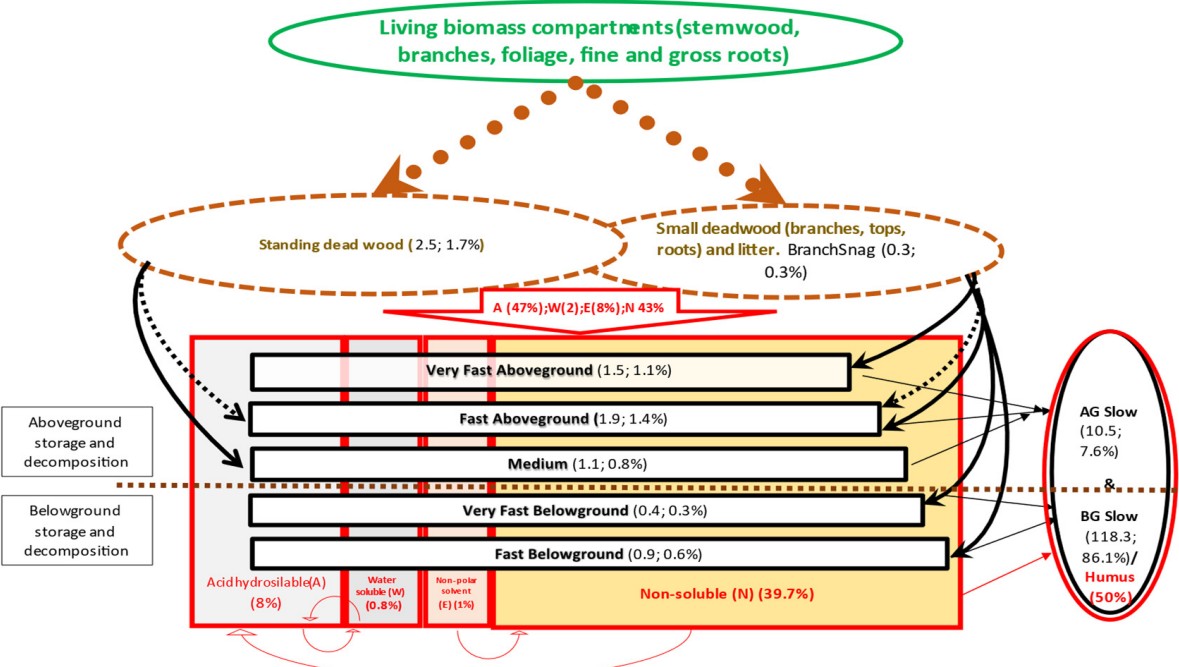

**Figure 1.** Overview of the conceptual frameworks and approximation of C stocks and flows for CBM and Yasso15. Note: Solid flows of carbon are only shown on the graphs (i.e., fluxes are not shown): biomass accumulation (in green) and transfers from living biomass to dead organic matter of the soils (in brown). C pools (as horizontal and vertical blocks) and transfers among them are shown in black for CBM and in red for Yasso15. Horizontal black blocks correspond to CBM C pools, while red vertical ones to Yasso15 soil C pools. Horizontal dotted line imitates the aboveground and belowground processes. Decay of fresh dead organic matter (continuous arrows) and physical transfers (dotted arrows) are shown. Thin arrows show transfers among the soils C subpools (to humus). C stock in living biomass (137.5 tC/ha, ABG 83.8%, BGB 16.1%). Living biomass turnover rates (% of standing stock): branches 2.7% softwood and 2.5% hardwood, coarse roots (stumps) 2%, fine roots 64.1% and for foliage 25% for softwood and 95% for hardwood species (excluding mineral components of biomass). Annual input to dead organic matter is shared between dead wood (DW, 8.1%) and litter (LT, 91.9%), annual transfer rate from standing DW to laying DW (8.8%). From total annual DW input, 40% is represented by stem wood and 60% by branches. C stocks are presented for each subpool in tC ha$^{-1}$ and in % as of total C stock in dead organic matter (100%). Values are estimated as average over simulated period in the BAU scenario. Pools and transfers are shown according to Kurz et al. [35] and Pilli et al. [33] for CBM and Tuomi et al. [45] for Yasso15.

Yasso15 simulates the decomposition of organic carbon by representing the state in five pools based on their solubility: Acid (A), Water (W), and Ethanol (E) soluble compounds as well as lignin based insoluble compounds (N) [30,46]. It is not autonomous on deriving inputs for which reason it is attached to other biomass models (e.g., EFISCEN, CO2fix). In addition to these four is the Humus (H) pool which contains stable, long-lived carbon compounds. The litter input can be also fractioned in four AWEN pools, thus connecting the carbon input directly with the state variables within the model. Dimensions of the litter inputs are dealt with by defining a threshold to discriminate between the coarse woody and the fine litter. The size of particles affects the decomposition speed: the larger the diameter is, the slower the AWEN pool decomposition rate will be. As carbon compounds are broken down in each pool, it is transformed to compounds belonging to the other pools or released to the atmosphere as $CO_2$. The decomposition rate is affected by soil temperature and moisture, with air temperature and precipitation used as indicator drivers here, as well as the size of the litter. Litter input is external to the model, so it can be

attached from any other model (e.g., CO2fix, EFISCEN) and the equilibrium will reflect the average litter fall described by the vegetation model.

### 2.2. National Forest Inventory

Romanian NFI1 (first cycle covers 2008–2012) records 6.98 mil. ha of forests [47] for the mid-year 2010, out of which 6.07 mil. ha represented forest available for wood supply (FAWS). In this study, we considered only FAWS. Within NFI-2 (second cycle covers 2013–2018) with the mid-year 2015, the reported forest area was 6.93 million ha. NFI records 73 tree species in Romanian forests. In FAWS, the most representative species is beech (31%), coniferous (26%) and oaks (16%). Only 22% of area is occupied by pure stands, 26% by two species stands, while rest have three to eight, or more, tree species. Their distribution within standing stock and increment shows similar shares. Overall, 53% of forest area is in age classes younger than 60 years, with the 2nd age class representing alone 21% of total area. According to NFI, forests show an average standing merchantable volume (excluding stumps) of 247.43 $m^3$ $ha^{-1}$, with an average current annual increment of the standing stock of 6.86 $m^3$ $ha^{-1}$ $yr^{-1}$.

### 2.3. Forest Soil Inventory

NFI soil data was available from national GHG inventory database [44]. Sampling forest soil organic carbon scheme was integrated within NFI sampling according NFI field data collection protocol [48]. Specifically, the sampling methodology considered the three "traditional" pools: organic matter of the mineral soil, litter and dead wood. Such classification should correspond to the pools defined by IPCC [14], and implemented by CBM [35]. A total of 5036 NFI plots were considered for mineral soil and dead wood analysis (one plot per cluster) on a 4 × 4 km grid in mountain and hilly areas and 2 × 2 km grid in plain area. Mineral soil (excluding litter) was sampled in pits on geometric horizons until 150 cm depth or the bedrock. Skeleton content was estimated in the field. The soil apparent density was extracted from digital maps available [49] and checked against existing national references on soil types [38]. Litter pool was sampled from the same plot as the soil, in a subset of 1158 NFI plots. For each sampling point, four samples were collected and processed individually throughout. Processing in the laboratory consisted in exclusion of biomass of non-woody grass and mineral residues through incineration. Dead wood volume was sampled as a regular NFI procedure [42], and conversion to C was done using standard wood density for the relevant tree species in the plot. All samples were collected in 2012 and 2013, so we assume 2013 as reference year for C stocks in soil. Lowest number of soils samples included in this analysis (*n* = 125, 2.4% of total samples) was available for *Robinia pseudoacacia* forests which cover some 250 thousand ha.

### 2.4. Litterfall Estimates

"Litterfall" is a generic term used here for the amount of living biomass transferred to forest floor, i.e., annual input of biomass to dead organic matter pool. Such transfers occur to either one of IPCC [14] carbon pools: dead wood as standing and lying with threshold diameter over 10 cm, and soil's litter pool containing non-woody, i.e., wood smaller than 10 cm in diameter, dead leaves and fine roots. The transfers from merchantable standing stock to dead wood pool were assumed larger than 10 cm in diameter (consistent with NFI definitions). Given data availability, only biomass from trees is included, so assuming that biomass from other vegetation types is negligible (e.g., understory).

For a harmonized initialization, simulation and validation of both models, FAWS was stratified for eight forest types across five climates (Table 1).

**Table 1.** The simulated forest types and the forest area (ha) in FAWS. In mixed forest, the proportion of each coniferous and broadleaved species is approximately 50%.

| Abbrev | Forest Types (The Share of Main Tree Species) | Area (ha) |
|---|---|---|
| FS | *Fagus sylvatica* (>90% beech) | 914,359 |
| PA | *Picea abies* (>90% Norway spruce) | 674,483 |
| QR | *Quercus* sp. (all oak species) | 505,508 |
| RP | *Robinia pseudoacacia* (black locust) | 123,069 |
| OB | Other broadleaved (>90% broadleaved species) | 2,668,032 |
| OC | Other coniferous (including *Abies alba*, silver fir, >90% coniferous species) | 32,861 |
| ConBroad | Mixed coniferous and broadleaved species | 527,284 |
| PreCon | Predominantly coniferous (>70% coniferous species) | 330,923 |

Forest status data is derived for NFI1, while all forest change parameters (e.g., increment, mortality) are derived from NFI1 and NFI2 [47]. Forest type characteristics like biomass allocation factors, species specific wood density (including for mixed forests types) and C content were implicitly captured in the CBM results on simulated carbon stocks or fluxes. Annual amount of litterfall is derived from CBM simulations and used by both the CBM as well as Yasso15 models for the initialization and simulation of soils C stocks for 50 years, a similar method was used by [50]. We assumed that our research questions would reasonably be addressed through analyzing the three selected scenarios for a short-term projection, i.e., only 50 years, rather than running period comparable to at least one rotation cycle. Thus, litterfall is derived for each type of biomass compartment from the simulations by CBM: merchantable wood (i.e., stemwood with bark), other wood (i.e., aboveground stumps and branches with bark), foliage, fine and coarse roots (diameter < 5 and >5 mm, respectively) according to [35].

Stands subject to silvicultural interventions experience litterfall also as residues resulting from harvesting operations. Their estimation is based on merchantability criteria (e.g., share of tops and stumps left as residues) and disturbance matrix defined for each type of disturbance. Stands without silvicultural interventions experience the transfers to dead organic matter as a result of the natural processes only. In order to estimate quantities of litterfall, CBM incorporates a turnover based solution for each biomass compartment. The analysis in this research included the most significant natural disturbance in Romania, the windstorms, with assumption that during simulated period annual events may occur within the range registered during 1990–2010 and that only 50% of biomass is removed by salvage logging compared to regular fellings.

Harmonization of litter input was performed for both initialization and simulations. Harmonization attempted mimicking the same input in Yasso15 as simulated by CBM, for both spin-off and actual simulation. By default, CBM implements internally a processing of the age-dependent and disturbance driven standing biomass dynamic on forest types, which cannot be extracted in that detail from the standard outputs. Consequently, the input to Yasso15 consisted in the average values corresponding to the most detailed stratification (Figure 1) extractable from CBM outputs (spatial grouping of climate, forest type and silvicultural interventions), where age is an implicit factor. Initialization consisted in determining the C stocks in the initial year of the simulation (i.e., 2013) corresponding to sampled NFI data. Validation was performed by comparing the total soil organic carbon initialized against NFI measured total soil carbon for the respective climate and forest type or their combinations.

*2.5. Harmonization of the Decomposition Process*

Romanian forests show a strong altitudinal stratification, with forests present up to 1700 m a.s.l., with the lowest temperatures and highest precipitations in high altitudes and

vice versa. In order to capture the vertical and spatial distribution of forests, NFI plots were allocated to five climatic units described by the multiannual averaged temperature and precipitation from ROCADA [51]. Consequently, the Romanian forests were associated to five climatic units with mean annual temperature ranging from 4.7 to 11 °C. Thus, climate consistent data, but appropriate to each model's requirements, was used (Table 2).

**Table 2.** Annual mean (Tm, °C), highest (Tmax, °C), and lowest (Tmin, °C) monthly temperature and annual precipitation (mm) for each climate unit (CLU) as input in CBM or Yasso15. Tamp (°C) represents the half of the difference between maximum and minimum monthly temperatures.

| CLU Code | Tm | Tmax | Tmin | Tamp | Precipitation |
|---|---|---|---|---|---|
| 44 | 4.7 | 19.3 | −9.6 | 14.4 | 886.3 |
| 35 | 6.7 | 22.0 | −8.4 | 15.2 | 823.1 |
| 34 | 8.3 | 24.2 | −7.4 | 15.8 | 751.7 |
| 26 | 9.8 | 26.2 | −5.7 | 15.9 | 748.7 |
| 25 | 11.0 | 27.7 | −4.6 | 16.2 | 678.2 |

*2.6. Scenarios*

Historical forest management practices and implicit harvest levels on forest types were retrieved from NFI2 and NFI1 database, so reflecting actual interventions rather than theoretical approaches from forestry guidelines. They were modeled in CBM as function of stand age and intensity of interventions. Since harvest has a significant impact on the litterfall amount, we performed simulations on three forest management scenarios: (1) business as usual (BAU) scenario where the annual harvest was approximately 60% of the volume increment or between 0.10 and 0.14% of the standing stock (ratios based on NFI's estimates), (2) no harvest scenario (noDist) which maximizes the biomass accumulation in the standing stock but also drives an increase in the mortality rate, which can be considered as the extreme case of "extensive" forest management practices, and (3) maximum intensity of silvicultural interventions (maxH) where the harvest volume equaled the annual biomass growth, which can be considered as the extreme case of "intensive" of forest management practices. Notably, there was a significant change in forest management in Romania over the last 50 years, which was not necessarily captured as modeling assumptions: the forest management was more systematic and intensive in the pre-1990 period compared to post-1990 [52].

*2.7. Data Processing*

Scenarios were run with annual time step until 2060. Harmonization of various databases regarding forest types (from NFI), climate (from ROCADA) and soil organic matter (from IFN) were processed in R, ArcGIS and MS Access. A Yasso15 version was run in R. The comparison of the models' performances was performed through analysis of residuals of simulated against measured data with the use of normalized root mean square error (NRMSE) as the relevant performance metric.

Additional analyses were only performed for CBM outputs as it allows a split of the total soil carbon on three sub-pools measured by NFI, whereas Yasso15 does not provide such a split. For such comparisons, it was assumed a CBM-NFI correspondence: soil organic matter (SOM) represents the C pool of stable organic matter in the mineral part of the soils which has turnover time of 300–500 years or even more while it also represents the largest share in the total stock in soils. Comparatively, litter and deadwood pools represent dead organic matter pools with turnovers generally between 1–3 and 5–20 years, respectively.

**3. Results**

*3.1. Litterfall Amounts during Spinoff*

The amount of the litter input to DOM used for the spinoff (initialization) varied by two or more orders of magnitude among the selected forest types, apparently closely reflecting the altitudinal distribution (Figure 2). To optimize the harmonization of the

initialization of both models, the data was extracted from results of CBM, run for 50 years at the lowest possible spatial disaggregation which potentially allowed representing the optimal approximation of input to DOM, i.e., intersection of forest type, climate, disturbance regime and criteria for stratification. Thus, a non-age-dependent input was used for Yasso15 spin-off.

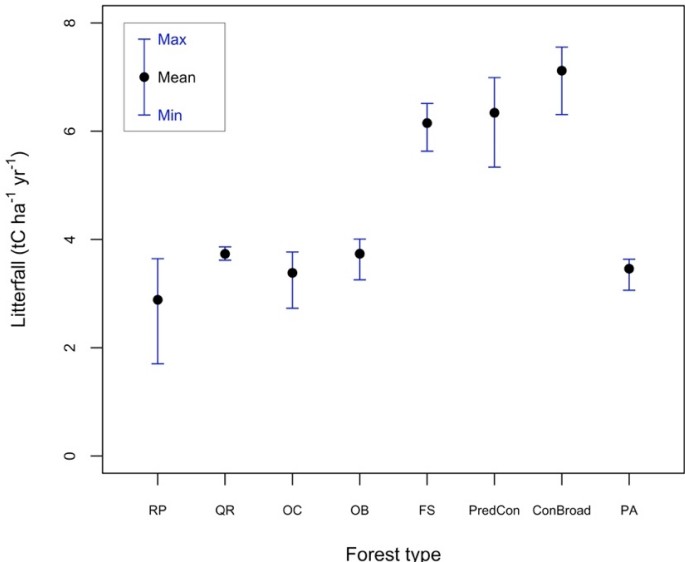

**Figure 2.** Amount of litterfall (tC ha$^{-1}$ yr$^{-1}$) used for the spin-off (initialization) for each forest type selected. Bars represent the maximum and the minimum values over the 50 years of CBM simulations assimilated to spin-off dataset. The abbreviations for forest types are shown in Table 1.

### 3.2. Model Performance for the Initialization of Total C Stock

Based on NFI database, total soil C stock was the highest in mountain forests, dominated by *Picea abies* (PA) and the lowest in lowland forests dominated by *Robinia pseudoacacia* (RP) and *Quercus* sp. (QR) (Figure 3). The altitudinal trend of increasing C stock is both recognized for forest types (Figure 3a) and climates (Figure 3b). On forest type, the variation coefficient ranged from 40% up to 125% depending on the forest type being lowest for PA and largest for RP.

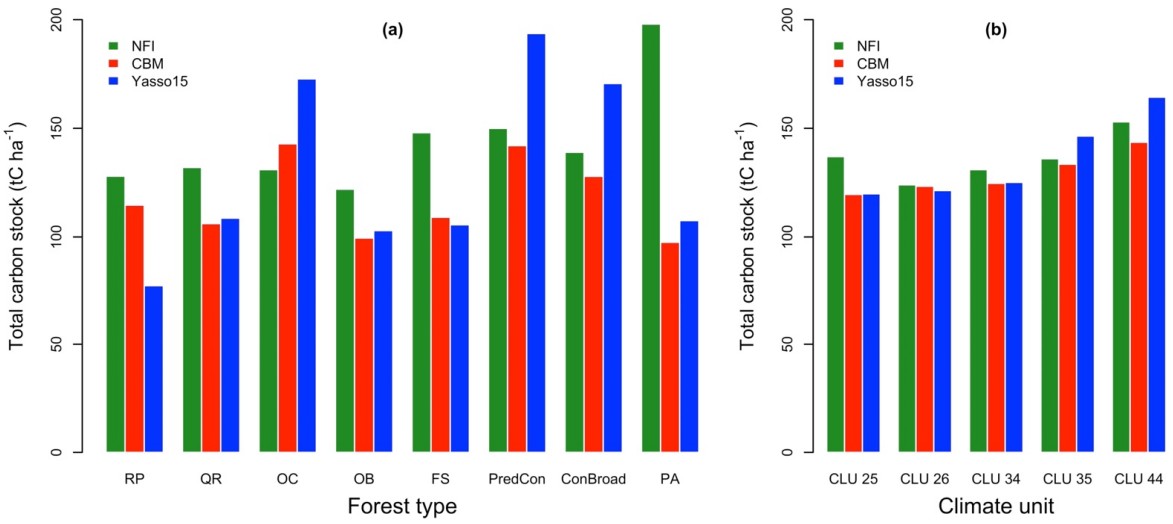

**Figure 3.** Measured and modeled initialized total C stock (tC ha$^{-1}$) on forest types (**a**) and on CLUs (**b**). Forest types and CLUs are represented following the altitude increase, from left to right. Decreasing number in the CLU indicates increase in mean annual temperature. See the abbreviations for the forest types in Table 1.

Both models tended to slightly underestimate the total C stock, especially when the stratification was on forest types only (Figure 3a) rather than on climate only (Figure 3b).

The NRMSE values of simulated vs. measured NFI data was approximately 26% (of the average C stock, for both models) for both, when stratification was done by forest types only and when stratification was done by climate and forest types. NRMSE reduced to 13% (for both models) when the analysis considered stratification on climate only. Still, CBM performed slightly better when analyzed as the absolute difference to the measured values, i.e., the differences were approximately 15% smaller than those of Yasso15. Figure 4 shows that simulated values for the projected period matched better for smaller C stocks rather than for higher values. For the upper range of C stock, there seems to be some overestimation by Yasso15, especially for mixed forests of coniferous and broadleaved species.

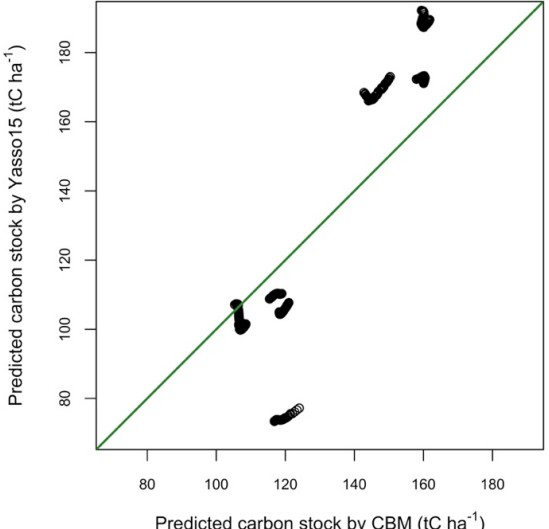

**Figure 4.** Reciprocal achievement of simulated values by the two models. Grouping of values on the graph is related to the stratification on forest types and climates. Green line represents 1:1 match.

Figure 5 shows the agreement (i.e., lower RMSE values) between NFI and both, CBM and Yasso15 estimates, when CLU (climatic unit) was used as a driver for the output representation. However, currently, the CLU has no practical value thus far, as the forestry sector rely on forest type stratification, rather than other criteria.

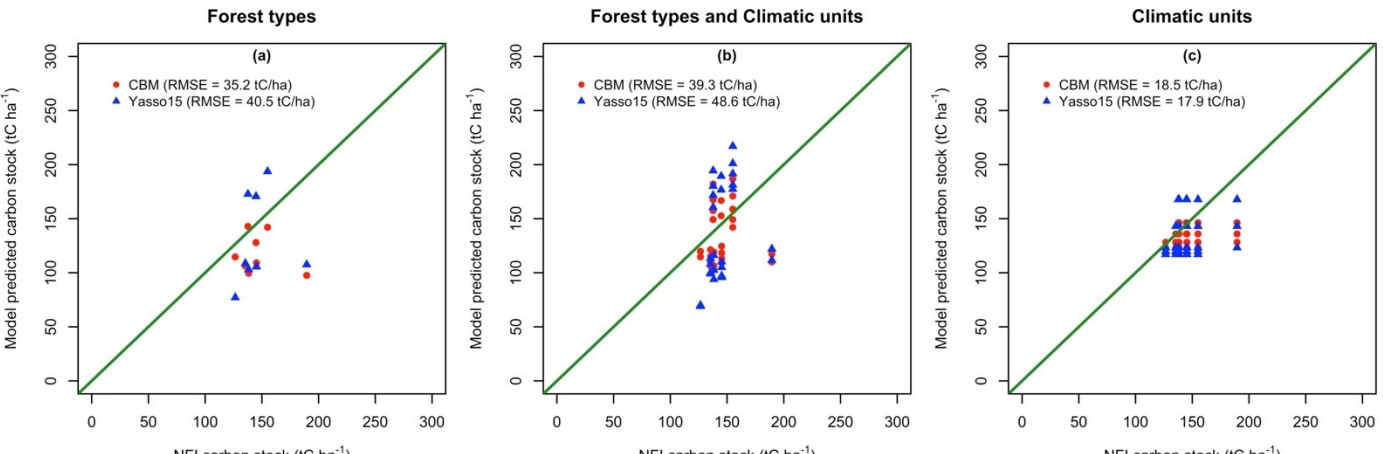

**Figure 5.** Measured and modeled C stock when the data is pooled on forest type (**a**), forest type and CLU (**b**), and CLU alone (**c**). Yasso15 is denoted with blue and CBM with red color. Green line represents 1:1 value on the OX and OY axes.

### 3.3. Initialization of C Stocks in the Soil Subpools

The share of SOM in total C stock was 85–90% and 95–98% by CBM simulations and for NFI measured data, respectively. Further on, for both litter and dead wood, CBM generally simulated within one to four order of magnitude smaller C stocks than measured ones (Table 3).

**Table 3.** Range of NFI measured (within 95% confidence interval of the mean) and initialized C stocks (tC ha$^{-1}$) in the subpools of CBM on forest types and CLUs. SOM = soil organic matter, LT = litter, DW = dead wood, Total = sum of the three subpools. Values without range represent an average of a small number of samples in the available data pool. Totals are rounded to the integer. See the abbreviations for forest types in Table 1.

| Source | Pool | PA | ConBroad | FS | QR | OC | OB | PredCon | RP |
|---|---|---|---|---|---|---|---|---|---|
| NFI | SOM | 131.1–195.3 | 103.9–149.2 | 113.1–158 | 101–158.8 | 89.9–139.3 | 117.6–169.9 | 131.4–138 | 120.5–129.9 |
| | LT | 8.1 | 4.5 | 4 | 2.9 | 5.2 | 1.6 | 4.7 | 2.2 |
| | DW | 0.6–1.6 | 0.7–1.7 | 0.5–1.2 | 0.1–0.4 | 0.2–2.4 | 0.1–2.3 | 0.5–2.1 | 0.2–1.1 |
| | Total | 135–205 | 108–155 | 117–163 | 104–162 | 95–145 | 119–172 | 137–143 | 123–128 |
| CBM | SOM | 88.4–100.8 | 124–153.7 | 126.7–151.9 | 96.4–106.3 | 90.4–103.2 | 100–110.3 | 113.8–139.9 | 104.7–111.3 |
| | LT | 5.7–11.6 | 14–28 | 11.8–23.6 | 7.2–11.7 | 5.7–11.5 | 6.5–10.8 | 11.5–23.2 | 5.6–8 |
| | DW | 3.4–4.9 | 4–5.1 | 4.4–6.5 | 2.6–3.4 | 3.4–4.7 | 2.6–3.5 | 2.7–3.6 | 4.3–5.5 |
| | Total | 97–117 | 142–187 | 143–182 | 106–121 | 99–119 | 109–125 | 128–167 | 115–125 |

Overall, the coefficient of variation of measured C stocks on forest type was 87% (43–132%) for SOM, 8% (6–23%) for litter and 368% (129–387%) for dead wood, which is on average some 174% for the total C stock. In the background calculation of SOM, the coefficient of variation for the C concentration in soil samples for SOM was only 27% (13–45%).

### 3.4. Litterfall Dynamic for the Scenarios

The amount of annual litterfall simulated by CBM was around 3% of the standing stock for the BAU scenario. Despite general comparable levels, the three scenarios showed particular trends (Figure 6). On average, when compared to spinoff values, the litterfall input was 15% and 41% higher for BAU and maxH, respectively, or equal for noDist scenario. The harmonization of litterfall between models resulted in 16% higher input in Yasso15.

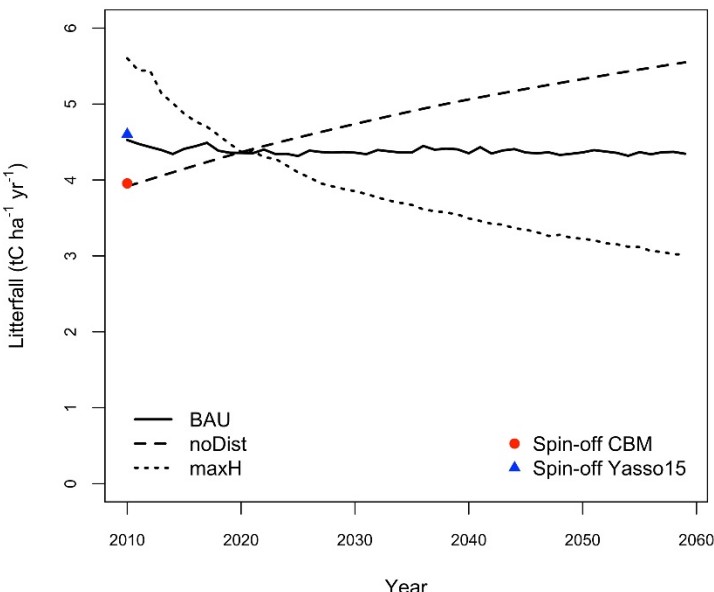

**Figure 6.** Trend of the litterfall averaged for all forest types, over the simulated period associated to the three scenarios: business as usual harvest scenario (BAU), no disturbances scenario (noDist), and maximum harvest scenario (maxH).

The noDist scenario was associated to a progressive increase of the litterfall, while the maxH scenario resulted in a decrease of litterfall. Litterfall input was notably higher for mixed coniferous-broadleaved (e.g., PredCon, ConBroad) and *Fagus sylvatica* forests compared to the pure species forests. There were also some peaks or steeps in litterfall on forest types over the projected period, but those were caused by the assumptions regarding the management or natural disturbances.

The largest contribution to litterfall was represented by foliage (Table 4). The woody components, i.e., merchantable and other woody, sourced from the aboveground biomass compartments represented some 20–24% of total litterfall. Fine and coarse roots contributed from 42% (for noDist scenario) to 52% (for maxH scenario), with BAU in between (45%). Different shares of contribution for spin-off and noDist by BAU and maxH scenarios is explained by the presence of silvicultural interventions, with the spin-off and noDist representing counterfactual situations where only processes simulating natural dynamics of biomass occur (i.e., mortality and other compartment turnovers).

**Table 4.** Contribution to litterfall from natural processes (i.e., consequence of natural turnovers of the biomass compartments) and from forest residues (from silvicultural operations) in the spin-off (initialization) and for the simulation of the three scenarios. NA applies for residues from forest operations in noDist scenario.

| Scenario | Litterfall Origin | Merchantable Standing Stock | Other Woody Compartments | Foliage | Fine Roots | Coarse Roots |
|---|---|---|---|---|---|---|
| Spin-off | Natural turnovers | 8 (2–17)% | 12 (4–25)% | 38 (3–58)% | 30 (19–52)% | 12 (7–22)% |
| BAU | Natural turnovers | 6 (2–11)% | 16 (4–40)% | 28 (2–55)% | 36 (17–50)% | 16 (9–33)% |
|  | Forest operations residues | 15% | 16% | 37% | 13% | 39% |
| maxH | Natural turnovers | 5 (2–13)% | 19 (1–48)% | 24 (1–50)% | 27 (14–53)% | 25 (12–52)% |
|  | Forest operations residues | 25% | 29% | 24% | 11% | 45% |
| noDist | Natural turnovers | 7 (2–11)% | 14 (4–24)% | 36 (7–59)% | 30 (20–52)% | 13 (7–21)% |
|  | Forest operations residues | NA | NA | NA | NA | NA |

Overall, woody amount is generally some three orders of magnitude lower than non-woody inputs, while the actual quantity strongly depended on silvicultural intensity interventions (Figure 7).

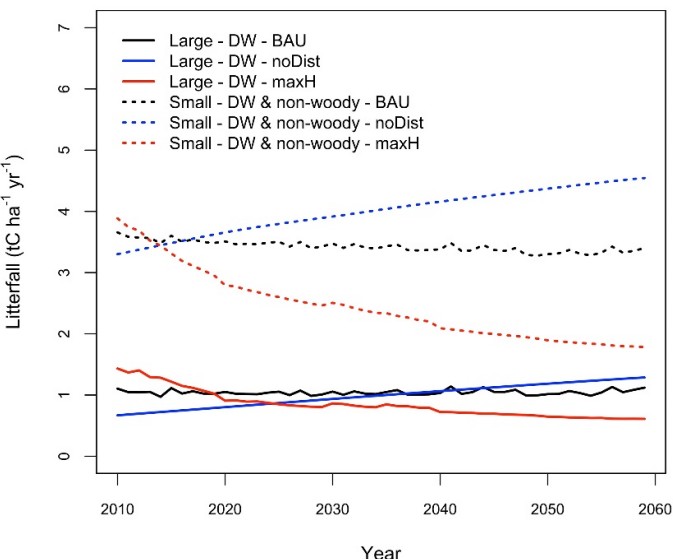

**Figure 7.** The litterfall dynamic split on large dead wood (diameter > 10 cm) and small-wood and non-woody components (diameter < 10 cm, foliage, fine roots) for the three scenarios.

The litterfall shares linked to silvicultural interventions were rather small. The large wood fractions, i.e., large dimensions of stem wood with commercially relevant dimensions, represented only 15% under BAU. For the other two scenarios, it was much smaller, given the total harvesting of available biomass (under maxH) or limited to the contribution from natural mortality (under noDist). In fact, the harvest-based demand led to silvicultural interventions on only approximately 4% of the total forest area annually. On average, it represented an area of 60 kha that was a subject to final cuts and 200 kha that was subject to thinning operations annually, which explains the low contribution of silvicultural interventions to total litter input.

*3.5. Projections of Soil Total C Stock and Dynamics of the Annual C Stock Change*

Generally, both models simulated a similar development in the total C stocks within different scenarios (Figure 8a). In BAU, there were very few changes over time (Figure 8b). Such a flat dynamic under BAU shows both consistent litterfall input and consistent decomposition for the simulated period compared to initialization (as shown in Figure 6). The C stock was the highest by the end of the simulation period in the noDist scenario. By opposition, maxH scenario showed the smallest one. The short time increase and, respectively, the decrease of C stocks in the first two decades simulated under maxH and noDist show primarily an unbalance of litterfall inputs between simulation compared to initialization.

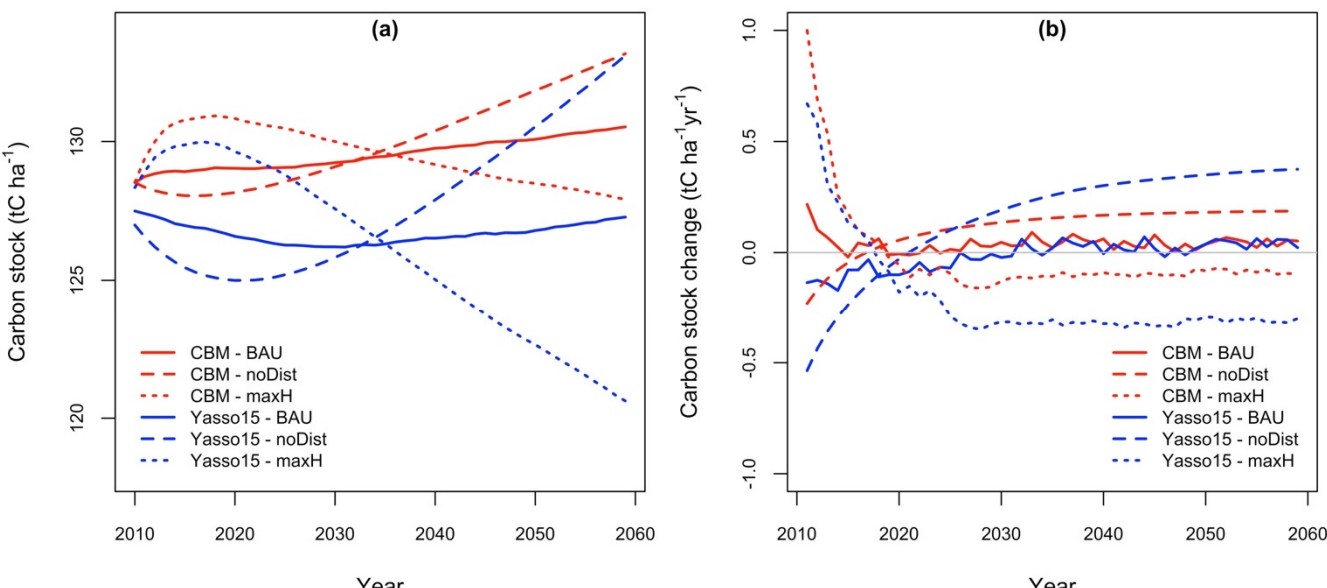

**Figure 8.** Total C stock in the dead organic matter pools, averaged for all forest types (**a**) and annual change (**b**) in total C stock, corresponding to the three scenarios. The values are averaged for all eight forest types.

Toward the end of the simulated period both models consistently converged toward an equilibrium where the soil acts as a small sink of approximately 0.05 tC yr$^{-1}$ for BAU. Despite overall similar trends between the models, there were slight deviations in the two counterfactual scenarios which result in roughly double loss of carbon by Yasso15 for noDist scenario and a double C gain by CBM for maxH scenario. With exception of BAU, both counterfactual scenarios also showed either convergence or divergence toward 2050, which is, most likely, related to decomposition given the unbalanced input to DOM in the initialization and each scenario. BAU was the most consistent one, as there were negligible differences of litterfall input in the initial simulated year.

The scenarios with simulations of silvicultural interventions demonstrated that total soil C storage was strongly affected. Specifically, both models projected decreasing C stocks for BAU and maxH, compared to noDist.



For the first 10 years of the simulations, the C stock changes were larger than for later period in all scenarios (Figure 8b), i.e., showing a start-up effect. In fact, the effect was stronger in the initial year and was decreasing sharply afterwards. Moreover, the largest initial effect was shown in the case of maxH scenario where litterfall inputs in the first simulated year is on average 25% higher (from 11 to 46% on forest types) compared to BAU's and spin-off, as well, in case of noDist scenario, it was 15% lower ($-6--30\%$ on forest types).

### 3.6. Simulated Soil Carbon Stock Change by Subpools

The SOM, that presented the slowest decomposition rate, showed negligible changes during simulation period, whereas the dead wood subpool showed the greatest change (Figure 9). Overall, there was a moderate, significant correlation across forest types (r = 0.25–0.35, $p < 0.05$) between litter input and annual C stock change of the fast-decomposing subpools simulated by CBM, when the simulated values for the first 10 years were excluded. For the slow decomposing subpool, the correlation was not significant ($p > 0.05$).

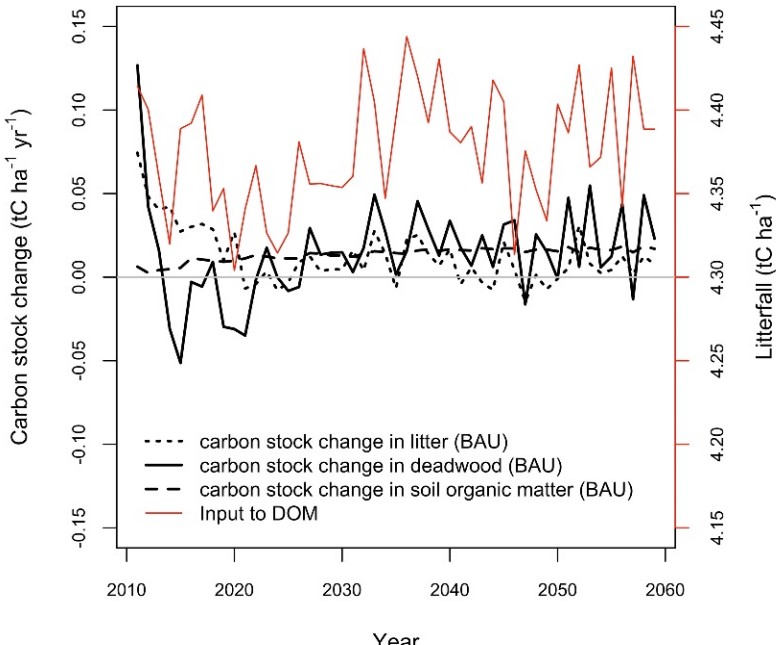

**Figure 9.** Simulated C stock change in each C subpool and litterfall by CBM for one climatic region for BAU scenario. The values presented are averaged for all eight forest types.

The C stock in SOM, which had the slowest decomposition rate as simulated by CBM, diverged negligibly from BAU along simulated period reaching at the end of simulation $-0.2\%$ for noDist scenario and $+0.5\%$ for maxH scenario.

## 4. Discussion

Romanian forests are compositionally diverse: 27% of forest area is based on single-tree species, while 46% of forests contain more than three species [42]. Empirical data from NFI shows that the more diverse forests contain a higher total C stock in the soils, which may be caused by the higher C stock in dead wood [53]. This is confirmed by our simulations (Figure 2) where mixed forests (predominantly coniferous and mixed between coniferous and broadleaved) showed the largest litter inputs and the largest soil C stocks (Figure 3). This comes in contradiction to findings that showed coniferous stands to have a greater capacity to sequester SOC compared to broadleaved forests [19]. Simulated C stocks varied noticeably among climate units for the same forest type (see SOM values in Table 3). This means that the spatial soil continuum is not recognized in these simulations because of our choice for a discrete stratification on forest types or/and climates. Data

was run on homogenous strata, i.e., representing large spatial areas, although running the models on granular, individual NFI plots, is recognized to produce better results for more robust C stocks and stock changes estimates [17,18,33]. Moreover, a single management approach is assumed for each type of forest, while in fact a range of approaches may occur in practice from the extensive to highly intensive. Low C stock for *Robinia pseudoacacia* (RP) and other coniferous (OC) forests was most likely related to their presence on poorest sites, which was represented in the litterfall data, as site productivity was not included in the biomass simulation. On the other side, both models operate with average environmental conditions and annual time step, which support the option of validation at the regional scale, rather than granular one [37]. Including environmental parameters, as well as better consideration of continuous and categorical features, on top of forest type specificity, improves the predictability of soil C stocks [54,55]. In opposition, models running based on average climate data, as in our case, omit extreme weather years, e.g., the impact of droughts to the decomposition of organic layers [36].

Matching the litter input to DOM in the two models was only partially achieved for the initialization. Despite harmonization efforts, we succeeded to run Yasso15 by averaged inputs on forest strata, while CBM ran at much more detailed level, i.e., stand age by time step of one year. The question remains whether such a simplified approach rendered Yasso15 less sensitive to time variation of the biomass inputs to DOM. Based on the results presented in Figure 3, we found no evidence that the strata averaged input vs. age-dependent input had a meaningful impact on the initialized amounts, since we did not observe any bias. This may be due to the large number of iterations achieved during the initialization, i.e., mimicking hundreds of years of interaction of litter inputs and decomposition of organic matter. When analyzed at national scale, the NRMSE values were practically similar when simulated values by each of the two models were compared to NFI data.

For the Romanian forests, the biomass input to soils in *Picea abies* (PA) forests was less than half of the amount in other forest types (to which *Picea abies* tree species contributes, like for coniferous broadleaved mixtures), while both models simulated similar C stock values for both initialization and simulated period. As far C stock estimated by NFI is accurate [38], it seems there was a failure to reasonably simulate either the living biomass compartments or the turnover rates that allows litterfall inputs. In our case, the underestimation of soil C stock in PA forests by both models, it is most likely linked to the amount of litterfall simulated, so further linked to the compartmentation of the living biomass and/or turnover rates for this forest type as implemented into the CBM. In fact, a recent intermodel comparison exercise with harmonizing input data for biomass (i.e., yield and growth, biomass expansion) showed difference in the initialized C stock in all C pools by CBM compared to IFN reference data (+6% for initialized standing biomass) and +30% more total C in the soil, attributable to data preprocessing as the input into CBM.

The total soil C stock seems rather realistically simulated in our study when looking to other studies. Dincă et al. [38] estimated similar stock values based on the soil parameters regularly sampled as part of the Romanian forest management planning (the majority of the forest experiences a planning every 10 years). In Hernández et al. [56], the country-wide averaged C stock was estimated as about 57 (27–82) tC ha$^{-1}$, which is rather half or less, compared to the NFI estimates in this study. Lower values in [56] are most likely explained by the underestimation of the litterfall input which was generated from forest management planning database which demonstrates less standing stock and less net annual increment of forest compared to NFI estimates [47,57]. This proves once again the importance of the accuracy of litterfall inputs in modeling realistic soil C stock. This refers further to the assumptions on contribution of non-woody litter from understory vegetation and turnovers of fine roots. On the other side, any variation in stands' horizontal and vertical structures is assumed fully captured by our empiric-based approach as of NFI data, e.g., reduction of average leaf area index, so is expected to be reflected implicitly within each scenario.

There is ample evidence about the effects of management on the amount of C stock in the organic layers of the forest floor, but there is much less information about measurable effects of management on stable C pools in the mineral soil [18]. Harvesting, particularly clear-cut harvesting, generally results in a reduction in soil C stocks, especially in the forest floor and upper mineral soil [20,58]. The cumulated effect at national scale depends on the extent of the land subject to management, i.e., a small area, some 9% of area is affected annually by silvicultural practices, and further on only 0.49% is actually subject to clear-cuts according to our BAU scenario (built on NFI data). However, the impact cumulates over time as shown by diverging trends of total C stocks by the three scenarios.

Slightly better match of the initialized C stocks by CBM to measured data may be explained by the "non-equilibrium" modeling approach by CBM compared to "equilibrium" approach by Yasso15 (see Figure 3). Indeed, CBM and Yasso15 have different initialization procedures and therefore initialize different moments for stands' age dynamics. According to Kurz et al. [35], CBM provides the C content in all C subpools in the initial year of the simulation (i.e., 2013) approaching a non-equilibrium soil condition. It does that by applying wildfires as a solution to saturate the soils C in the 'slow' subpool. 'Slow' subpool is composed of aboveground DOM (i.e., F, H and O horizons) and belowground DOM (e.g., humified organic matter in the mineral soil). Wildfire disturbance means that the living biomass and other subpools like litter and dead wood are fully burned, from every few decades to few hundreds of years, under specific parametrization of user-defined stand-replacing fires. Therefore, a significant weight is given to 'SOM slow' subpool saturation over the ephemerous pools corresponding to the rest of litter and dead wood pools which have half-lives more than 10-times smaller than SOM pool. Thus the 'slow' pool amount is stabilized based on the 1% convergence of the 'slow' subpool. However, there is an additional step performed to complete the initialization, which consists of further 10 repeated cycles without natural disturbances and one last user-defined management disturbance (i.e., clear cut in our case) before litter input from growing stand to the age recorded in the forest inventory. This way, CBM ensures the SOM stabilization which is indeed less prone to short term impacts like disturbances.

Another explanation for the difference between simulated and measured values may be the missing input from understorey vegetation in the simulation of biomass. As local data is not available in our case, we assumed to have a negligible contribution to litterfall, despite studies showing that litter input from understorey may be significant, e.g., as in northern Finland [36]. Even NFI data shows presence of bushes in Romanian forests on more than 50% of total forest area [42], but no quantitative measurement are performed on bushes.

Particularly in the case of counterfactual (maxH, noDist) scenarios, there is a "startup" or "coldstart" effect, i.e., over the first 10 years of the simulations (Figure 8b). Most likely that arises from the quantitative difference between litterfall input to DOM in the first years of the actual simulation and that of the initialization.

Our simulation showed a very small increase in SOM in the long run under the BAU scenario. Although, metadata research shows that the long-term impacts of forest managers' decisions on soil organic carbon (SOC) remain unclear given restructuring of soils C on soil profile [59]. Harvesting level though shows a clear impact on both litter input and dead organic matter dynamics. Extreme and counterfactual scenarios, noDist and maxH, lead to significant levels of change and opposite trends in time. No intervention assumption (noDist scenario) results in a reduced input early in the simulation period which increases back later through accumulation mostly due to increasing fraction of dead wood (+12% total input to soils compared to BAU while large wood mortality remains around 8% from total input). In contrast, when harvesting the entire amount available by maxH results in a high accumulation during first years of simulation given the high inputs from silvicultural operations. On the other side, when looking into SOM dynamics simulated by CBM under noDist (extensive management) and maxH (intensive management) scenarios, it seems the SOM stock dynamic does not confirm the metadata analysis which shows that harvesting

the residues would result in medium duration of C loss [20]. In opposition, no disturbance scenario results in a negligible loss of C from total soil pool. However, Jonard et al. [19] suggest further studies are required to elaborate forest management guidelines, so helping GHG management and forestry adaptation, i.e., climate smart forestry measures [60].

For Romania, the only available C stock change estimates were simulated by Yasso07 [56], showing a country-wide average gain of 0.05 MgC ha$^{-1}$ yr$^{-1}$, with a variation from a gain of 0.14 MgC ha$^{-1}$ yr$^{-1}$ for hardwood forests to a loss of 0.01 MgC ha$^{-1}$ yr$^{-1}$ for softwood forests. Under BAU, our simulated values stabilize long term at similar level. They are also comparable to the gain of 0.12 Mg C ha$^{-1}$ yr$^{-1}$ simulated for Finland [30]. Depending on harvesting particularities, larger gains are reported for Germany, either simulated by Yasso15 of +0.25 ± 0.10 Mg C ha$^{-1}$ yr$^{-1}$ or measured of +0.39 (±0.11) Mg C ha$^{-1}$ yr$^{-1}$ [37]. For France it is reported a gain of +0.35 Mg C ha$^{-1}$ yr$^{-1}$ based on repeated measurements [19] or a gain simulated by Yasso07 +0.45 (±0.09) tC ha$^{-1}$ yr$^{-1}$ vs. observed of +0.34 (±0.06) tC ha$^{-1}$ yr$^{-1}$ in a soil survey [61].

The performance of the models depends, first of all, on the adequate estimation of the litter inputs and model parametrization. The decomposition parameters between the two models could not be harmonized as CBM runs the decomposition of physical C subpools, while the Yasso15 runs decomposition on biochemical compounds. Nevertheless, given low values of NRMSE, the default parametrization of each model we tested here seems to provide an acceptable solution for simulations of C stock and C stock change when stratification is performed by forest type and climate.

Each model has its own particularities: Yasso15 provides estimates of soil aggregated pools, while it is very flexible in using localized data (e.g., at NFI plot). The CBM version used here implemented a unique set of decomposition parameters across all strata (e.g., climate units and forest types). Although not confirmed in this study, this may make the model less flexible in simulating C stock across smaller areas or territories with large combinations of climates. Parameters involved in decomposition equations and transfers between pools may not fully reflect the climate variation in Romania, especially for dead wood and litter, for all forest types (despite overall good match) when strata instead of plots are considered.

According to NFI measured data, SOM represents the largest share of total SOC stock (>95%), a result that could not be reproduced by CBM. Despite clear definition and understanding of the three soil C pools, it remains very complex to parametrize and validate against measured values, while avoiding double-counting of litter layers or missing parts of the sample which can lead to underestimation. Nevertheless, with all these in mind, we expect that the total C stock is not underestimated given the actual method implemented in sampling all C subpools on the ground by the NFI (i.e., where parts not sampled in one subpool are sampled in another). As response to such complex reality, Yasso15 reports a total C stock, and the split on subpools (e.g., like IPCC pools) is not possible without making additional assumptions and simplifications on the results. Overall, methodological shortcomings and knowledge gaps affecting soil studies may be strengthened by simultaneous use of multiple models [62].

NFI calculated C stocks also show inherent uncertainty. Total C stock estimates showed a high coefficient of variation (i.e., 174% on average) compared to C concentration in the mineral parts of the soils (i.e., 27%). This may be linked to the stratification on forest types, while variation may be expected to be lower on soil types. Additionally, NFI only collected data on C concentration and skeleton content while soil apparent density was obtained from model-based procedure [49], being known that soil density has a significant influence on the C stock [24]. Scarce and nonsystematically sampled data on soil apparent density exists nevertheless in Romania, having the same order of magnitude as those used here [38,63].

## 5. Conclusions

The two models performed satisfactorily in predicting soil C dynamics under harmonized climatic and litterfall input, despite their totally different modeling principles. The default decomposition parametrization seems to provide an acceptable solution for simulations of soil C stocks when forest type strata are combined with climate units. Both models showed similar performance for the forests with both high and low C stocks for the mineral soils sampled in NFI. Regional/local scale, as the alternative to national one, represents a reasonable spatial area for the validation of soil modeling outputs against empiric NFI data.

The availability of measured soil data for only one moment in time supports the initialization and simulation at regional scale. A methodological challenge related to "forward" calibration, i.e., assimilation of new data, increases, as repeated data from successive soil monitoring becomes more often available. Limitations of the models are mostly related to availability of data for understorey vegetation, data for living biomass and turnover rates in standing forests.

Results of C stocks and C stock changes can be taken into account for the reporting of the national GHG inventory of Romania, including for the demonstration that forest soils do not represent a net source of emissions given current mix of forest management practices. Results show that increasing the management intensity through more intense silvicultural interventions most likely results in small losses from total soil C stock, or contrary to small to negligible increases of C stocks when harvest is significantly reduced.

**Author Contributions:** Conception and design of study by V.N.B.B.; J.L.; L.K.; T.V. and M.M.; acquisition of data by G.M., G.G.; analysis and/or interpretation of data by V.N.B.B., L.K., T.V., I.D. and M.M.; drafting the manuscript (V.N.B.B., L.K., T.V.) and revising the manuscript critically for important intellectual content (L.K., I.D.). All authors have read and agreed to the published version of the manuscript.

**Funding:** The research associated to this article was developed as part of contract 88/2014 with Ministry of Environment (Romania) and project ERA-GAS/ERA NET's FORCLIMIT contract 82/2017 by UEFISCDI Romania. Academy of Finland (grant numbers 297350, 277623) and ERA-NET FACCE ERA-GAS project FORCLIMIT are acknowledged for financial support which has received funding from the European Union's Horizon 2020 research and innovation programme under grant agreement No 696356.

**Institutional Review Board Statement:** Not applicable.

**Informed Consent Statement:** Not applicable.

**Acknowledgments:** We thank the anonymous reviewers whose comments and suggestions helped improving the manuscript.

**Conflicts of Interest:** The authors declare no conflict of interest.

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
