# Peer review of "Silvicultural Interventions Drive the Changes in Soil Organic Carbon in Romanian Forests According to Two Model Simulations"

_forests, doi:10.3390/f12060795_

Round 1

Reviewer 1 Report

This is a good paper, well written and with interesting results. The methods as well as the results and the discussion (except one sentence) are fine. My main concern is however that the Conclusions seem too optimistic for me.

The first sentence in the Conclusions :“The two models performed well in predicting soil C dynamics and litterfall input, despite their totally different modelling principles” is not justified: Whether Fig. 4 a, nor Fig. 6 show good consistency with the NFI Results, which are the most realistic and data based ones. The overall mean (on a national scale) seems acceptable, the results however do not at all correlate. The results of the two models correlate with each other but none of the two models correlates with the NFI-results. This should be admitted, explained and in the Conclusions clearly stated.

There are two more points, which could improve the paper:

Table 4 is not clear to me:

What is a “Litterfall source” “Litter”?

Why are the sums of the percentages for the “Litter” always near 100 (which should be expected) but not so the sums of the “Forest Management”. For the noDist Forest Management I think it would be better to say that the percentages are not applicable, because the sum itself is zero, and thus the percentages are 0/0 which is undefined and not zero. Please explain better what “Litter” as ”Litter source” means – and not only in the text but also find a better way to say it in the Figure header.

In the title of Fig. 7 say : annual litter fall input for large woody (wood > 10 cm) ….

In line 346 the sentence, saying that Fig. 5 shows a better achievement seems wrong to me. See above for the lacking correlation. If you only want to show that the overall means better fit, a graph with the means and the confidence intervals would be a better depiction of what you want to say. Anyhow you will have to explain the lacking correlations and why this is not important in the context of your investigation.

Author Response

Reviewer 1.

This is a good paper, well written and with interesting results. The methods as well as the results and the discussion (except one sentence) are fine. My main concern is however that the Conclusions seem too optimistic for me.

Comment 1.1: The first sentence in the Conclusions: “The two models performed well in predicting soil C dynamics and litterfall input, despite their totally different modelling principles” is not justified: Whether Fig. 4 a, nor Fig. 6 show good consistency with the NFI Results, which are the most realistic and data based ones. The overall mean (on a national scale) seems acceptable, the results however do not at all correlate. The results of the two models correlate with each other but none of the two models correlates with the NFI-results. This should be admitted, explained and in the Conclusions clearly stated.

Response 1.1: We acknowledge the Reviewer’s concern and we improved the formulation, using “satisfactory” instead of “performed well”. Using “satisfactory” is supported by the low NRMSE values.

Further on, there is a weak correlation in case of forest type, also well noted by the Reviewer. However, there is a slightly stronger correlation of C stock when analysis is done on climate units only, so, when ignoring the stratification on forest types.

The poor correlation is explained by running the models on strata and not plot-by-plot (because of missing availability of NFI plot-by-plot data), additionally to general poor correlations when we talk on soils. In the background database we have some 5036 NFI soil samples, while modelling was run on average inputs from a combination of eight types of forests combined with 5 climate types. So, on average some 26 observations (5036 NFI plots / 40 (8 forest type x 5 climates x 5 regions) would correspond to one simulated value by each model. This results in no variability in case of simulated value compared to NFI plots corresponding to same (most detailed) strata, which yields weak correlation. From this perspective (of aggregated numbers), we can report a “satisfactory” functioning of the models. Overall, models are well recognized to work better when applied to large scale than on site level (e.g. Didion et al., 2017).

Comment 1.2: There are two more points, which could improve the paper: Table 4 is not clear to me: What is a “Litterfall source” “Litter”?

Response 1.2: Fixed, please see Table 4.  This was to show the inputs associated to silvicultural interventions (forest operation residues) and to natural growth and processes (natural turnovers, where there are no silvicultural operations in the respective year).

Comment 1.3: Why are the sums of the percentages for the “Litter” always near 100 (which should be expected) but not so the sums of the “Forest Management”. For the noDist Forest Management I think it would be better to say that the percentages are not applicable, because the sum itself is zero, and thus the percentages are 0/0 which is undefined and not zero. Please explain better what “Litter” as ”Litter source” means – and not only in the text but also find a better way to say it in the Figure header.

Response 1.3: Fixed, please see Table 4. Sums represent the average values across the respective range, indeed they match 100% at disaggregated level, e.g., for one year and one detailed strata, but when average over long term and across entire forest area they may not fully match 100%. We modified them, so they match 100% in the table, as the average percentage is only shown to demonstrate the simulation realism. 

Comment 1.4: In the title of Fig. 7 say: annual litter fall input for large woody (wood > 10 cm) ….

Response 1.4: We reformulated the caption of Fig. 7. “Wood diameter” is used instead.

Comment 1.5: In line 346 the sentence, saying that Fig. 5 shows a better achievement seems wrong to me. See above for the lacking correlation. If you only want to show that the overall means better fit, a graph with the means and the confidence intervals would be a better depiction of what you want to say. Anyhow you will have to explain the lacking correlations and why this is not important in the context of your investigation.

Response 1.5: Please see response 1.1 above, as well. The text was reformulated to convey a neutral message. The three graphs show the overall effect when introducing climate as variable.

Reviewer 2 Report

Hello authors,

As a scientist without subject matter expertise in soil organic carbon modelling I appreciate the care taken to thoroughly explain differences in CBM versus Yasso15. The comparison of carbon stock by climate unit and forest types estimated by each model versus NFI data was helpful in presenting compelling evidence that the models generate appropriate ranges of values for the soil substrate of interest.

The Results section was clear and thorough in presenting data and C stock accumulation trends across tested scenarios. There is valuable insight in your finding that a mixed composition forest can lead to increased SOC values, which has broad implications for re-forestation decisions in management of forests (shifting or maintaining community composition especially in the context of climate change). I feel that this paper has great utility if published to audiences of persons interested in seeing a thorough, systematic comparison of SOC model performance, and the applications of this type of analysis.

MINOR CORRECTIONS: Your reference managing software seemed to generate errors at the following line numbers - 221, 293, 361, 388, 476

Author Response

Reviewer 2.

Comment 2.1: As a scientist without subject matter expertise in soil organic carbon modelling I appreciate the care taken to thoroughly explain differences in CBM versus Yasso15. The comparison of carbon stock by climate unit and forest types estimated by each model versus NFI data was helpful in presenting compelling evidence that the models generate appropriate ranges of values for the soil substrate of interest.

The Results section was clear and thorough in presenting data and C stock accumulation trends across tested scenarios. There is valuable insight in your finding that a mixed composition forest can lead to increased SOC values, which has broad implications for re-forestation decisions in management of forests (shifting or maintaining community composition especially in the context of climate change). I feel that this paper has great utility if published to audiences of persons interested in seeing a thorough, systematic comparison of SOC model performance, and the applications of this type of analysis.

Response 2.2: We thank the reviewer for the positive feedback. Indeed, there is a very high number of results behind these simulations, which we did not show in the paper. Although, in early stage of the project we performed detailed calibration and validation exercise (e.g., on forest types against measured NFI data).

Comment 2.3: MINOR CORRECTIONS: Your reference managing software seemed to generate errors at the following line numbers - 221, 293, 361, 388, 476

Response 2.3: We thank the reviewer for suggesting these corrections. We corrected the manuscript.

Reviewer 3 Report

The manuscript covers insufficiently investigated area of carbon turnover and, actually, reveals methodological shortcomings and knowledge gaps affecting such studies. The article leaves several options for misinterpretation of results, therefore it should be improved to avoid extrapolation of the results beyond the study frame, e.g. interpretation of 50 years modelling results as long term effect, which is actually a bit more than a half of average forest rotation.

Recommended improvements:

  • lines 27-29 and beyond; The text should be supplemented with notation that this reflects only soil carbon stock, but not the total carbon turnover, most probably resulting in completely opposite values due to substitution effect and removals in harvested wood products. And 50 years is too short period to compare management scenarios. Basically the study compares harvesting and abandonment scenarios, which should be noted in the article. Reliable comparison should involve at least 2 rotations of trees, so that impact of regeneration and silvicultural measures are reflected. These shortcomings in the applied methods should be clearly state in abstract, methodology and discussion. At the same time the selection of such a short and non-reliable modelling period should be substantiated;
  • lines 102-104; it should be noted that Yasso can also model decomposition of coarse dead wood, which is actually important to acquire accurate projections;
  • lines 120-121; as already mentioned above 50 years modelling period is too short to answer this question. Probably this can be reformulated to short term impact, clearly stating that in long term the effect can be opposite, as many factors affecting the long term impact are not reflected in a period shorter than a single rotation of trees;
  • lines 129-130 and beyond; this is not fully correct, both models can't be used with organic soils and soils with high groundwater level. It is not clear, how shrubs, herbaceous vegetation, mosses and lichens are considered in the modelling. These carbon sources have particular importance during forest regeneration period. It is also unclear, how harvesting residues, stumps and roots of extracted trees are considered in the calculation. These sources also have significant impact on soil carbon stock, actually significantly increasing carbon input in soil directly after harvesting;
  • line 142; these values of deadwood and litter proportion demonstrates very healthy forests, which looks not realistic for over-mature forests in the extensive management scenario. Source of information should be added, as well as carbon sources included in litter pool;
  • line 147 on CBM model; it should be elaborated further, if the model can compare harvesting and abandonment scenario, considering that significant proportion of input in harvesting scenario comes from harvesting residues. Im not sure if the calculation approach implemented in CBM can calculate turnover of this carbon source in correct way, especially if such a short projections period is used;
  • line 178 on Yasso model; the size of litter can be elaborated further, how it is applied in the study, particularly, comparing harvesting and extensive management scenario;
  • lines 209-210; if I understand correct that the same density is applied to all kind of dead wood, this is leading to significant overestimation of carbon stock in dead wood, because density of dead wood decreases with age and carbon stock in partly decomposed dead wood is significantly smaller. This should be clarified;
  • lines 221-222; reference to Table 1 should be corrected;
  • chapter 2.4 generally; is it really so that non-woody litter is not considered in the calculation? It is also not clearly stated, how carbon input due to turnover of fine roots is considered;
  • line 277; how harvests are associated with increment? This statement should be changed to characterize proportion of harvested stands in comparison to stands / stock available for harvesting. Comparison of harvests and increment is useless information in projections of harvest rate;
  • lines 280-282; The need for theoretical ranging of harvest projections is understandable, but how harvest projections are associated with increment? Maximum harvest projections should be associated with with availability of wood in mature stands and wood available for extraction in thinning. If there is no legal association between permitted harvest rate and increment, this scenario should be renamed to avoid misinterpretation of the assumptions, otherwise reference to the legal act should be added;
  • lines 325-326; it is not clearly stated, how stand types effect is considered in calculation and how it is avoided in climate stratification scenario. The term "sligtly" is too optimistic here, there is significant and systematic underestimation;
  • line 361; reference should be updated;
  • Table 3; carbon stock in dead wood seems to be very small. This corresponds to 3-4 m3 ha-1. Probably this is the reason for underestimation of carbon stock in the modelling exercise? If possible this value should be verified by other studies;
  • line 388; reference should be updated;
  • line 476; reference should be updated;
  • lines 526-527; it is also reasonable to mention in discussion that impact of non-woody litter and turnover of fine-roots may be underestimated in the calculation, particularly in scenario considering reduction of average leaf area index of woody vegetation due to harvests;
  • lines 530-531; such conclusions should be based on long-term observations / projections. Is it so that forests in central and southern Sweden doesn't contain carbon in soil after 800 years of clear-felling history? Both sides of the story should be reflected in the discussion;
  • lines 649-651; this conclusion is not substantiated by the study due to short period of projections. At least 2 generations of trees should be calculated to see real long term effect. The formulation "long term" in conclusions should be changed to particular calculation period.

After proposed corrections and clarifications the article will become a valuable source of information for scientific community and politicians.

Author Response

Dear Reviewer, 

thank you for your effort to understand our paper, and above all to provide us with such reasonable, clear, and very constructive and useful comments. We implemented all of them. 

Reviewer #3

The manuscript covers insufficiently investigated area of carbon turnover and, actually, reveals methodological shortcomings and knowledge gaps affecting such studies. The article leaves several options for misinterpretation of results, therefore it should be improved to avoid extrapolation of the results beyond the study frame, e.g. interpretation of 50 years modelling results as long term effect, which is actually a bit more than a half of average forest rotation.

Response 3.0.a –Reviewer raises a very interesting scientific question, indeed. Nevertheless, our study is limited only to the decomposition of dead organic matter that occurs in forest, on forest land, and not the entire turnover and decomposition associated to forest and wood products. So, it does not intend to assess the entire decomposition of wood leaving forest sites. The focus on forest soil is justified as policy relevant issues and comparability of the two models. Text is added/improved on lines: among other places in the text, lines 114-123 clearly mention soil as subject of the study.

Response 3.0.b In the revised version a double check of the text and information transparency related to turnovers (i.e. from living biomass to dead pools, and among the dead organic matter sub-pools) and decomposition of organic matter, according to the models’ requirements are transparently explained and highlighted. Text is added/improved: line 638-640 addedd a short discussion issuing from the study are added to cover the methodological shortcomings and knowledge gaps affecting such studies.

Response 3.0.c - Indeed 50 years represents the half of the cycle duration for majority of European forest stands. Reviewer notes that having only 50 years as projection period is too short for soils. We decided to 50 years only so assuming that implementing three scenarios involved would provide for an adequate picture of models behaviour while avoid more assumption of the change of decomposition/decay under uncertain climate change. Additionally, 50 years is a policy relevant timeframe. Text is added/improved: justification of 50 years is added under the Table 1, lines 241-244.

Recommended improvements:

Comment 3.1: lines 27-29 and beyond; The text should be supplemented with notation that this reflects only soil carbon stock, but not the total carbon turnover, most probably resulting in completely opposite values due to substitution effect and removals in harvested wood products.

Response 3.1: We agree with the Reviewer. The study only covers the decomposition of dead organic matter that occurs in forest, on forest land. HWP and substitution effects, i.e., beyond forest lands, are not covered in this study, although the reviewer’s view on a total turnover provides for a very interesting research question, in fact. Text is added/improved on lines: text re-checked and clarified.

Comment 3.2: And 50 years is too short period to compare management scenarios. Basically the study compares harvesting and abandonment scenarios, which should be noted in the article. Reliable comparison should involve at least 2 rotations of trees, so that impact of regeneration and silvicultural measures are reflected. These shortcomings in the applied methods should be clearly state in abstract, methodology and discussion. At the same time the selection of such a short and non-reliable modelling period should be substantiated;

Response 3.2: We agree that 50 years may be short compared to the full length of a forest rotation, and so much less when talking on soils. However, the long-term projections are usually heavily affected by systematic errors that may build up year by year, i.e., systematic errors are propagateded and have uncontrollable effects for longer projections.

Another reason result from a major shortcoming in doing such simulations was harvest optimization as input into CBM. Going beyond 50 years would have resulted in unrealistic age class structure (under constant harvest scenario, CBM simulates accumulation of stands under old age classes, which is not realistic assuming a normalization of forest structure applied throughout Europe, as well as in Romania).

On the other side, a long-term simulation was replaced by simulation of two additional scenarios.

Another reason for limiting the simulation to 50 years was to inform policy makers on short term emission reduction possibilities, while understanding the robustness and reliability of soil modelling approaches. For this reason, we considered the impact of extreme management options over C stocks in dead organic matter pools. Text is added/improved on lines: Text on justification of 50 years is added under the Table 1.

Comment 3.3: lines 102-104; it should be noted that Yasso can also model decomposition of coarse dead wood, which is actually important to acquire accurate projections;

Response 3.3: We agree with the Reviewer. Yasso15 simulates decomposition of entire dead wood pool on two sub-pools with the threshold of 10 cm in our study. A new sentence was added in description of Yasso15 (line 178-171).

Comment 3.4: lines 120-121; as already mentioned above 50 years modelling period is too short to answer this question. Probably this can be reformulated to short term impact, clearly stating that in long term the effect can be opposite, as many factors affecting the long-term impact are not reflected in a period shorter than a single rotation of trees;

Response 3.4: Review’s point is clear, and we justified the reason for our selection of simulation of short term approach. Text is added/improved on lines: we agree, short term impact over the following 50 years is highlighted throughout the text. Although, we do not expect an opposite impact on longer term, but rather the trend shown in this study. At least in case of noDist scenario, soil may behave as sink for very long period of times (e.g., Nabuurs et al., 2013). Indeed, there is a large uncertainty in case of maxH scenario, but even there respecting the silvicultural prescriptions the C loss can continue to small levels.

Comment 3.5: lines 129-130 and beyond; this is not fully correct, both models can't be used with organic soils and soils with high groundwater level.

Response to comment 3.5: We agree that the sentence is confusing. We reformulated the sentence to refer to mineral soils only.

Comment 3.6: It is not clear, how shrubs, herbaceous vegetation, mosses and lichens are considered in the modelling. These carbon sources have particular importance during forest regeneration period. It is also unclear, how harvesting residues, stumps and roots of extracted trees are considered in the calculation. These sources also have significant impact on soil carbon stock, actually significantly increasing carbon input in soil directly after harvesting;

Response 3.6: Shrubs, herbaceous vegetation, mosses and lichens were explicitly excluded from simulation, due to the lack of data from NFI. This is deal in background CBM modelling where their biomass and turnovers are assumed nil (i.e. no expansion factors for sub-merchantable trees and vegetation). Text is added/improved on lines: explanation is added when defining the “Litterfall”, line 220.

Comment 3.7: line 142; these values of deadwood and litter proportion demonstrates very healthy forests, which looks not realistic for over-mature forests in the extensive management scenario. Source of information should be added, as well as carbon sources included in litter pool;

Response 3.7: The caption in figure 1 reflect the national averaged data from National Forest Inventory (www.roifn.ro). Just note that the average standing stock in Romanian forest is 340 m3/ha (to which adds C stock in belowground, foliage in order to get the total C stock in living biomass). Numbers in this caption are estimated by CBM, calibrated against NFI in the initial year of the simulation.

Comment 3.8: line 147 on CBM model; it should be elaborated further, if the model can compare harvesting and abandonment scenario, considering that significant proportion of input in harvesting scenario comes from harvesting residues. Im not sure if the calculation approach implemented in CBM can calculate turnover of this carbon source in correct way, especially if such a short projections period is used;

Response 3.8: Valid point, thank you. Our thinking is that on short term of 50 years the turnovers implemented in the modelling for all biomass components are still realistically represented for both managed and abandoned (noDist) scenarios. National average of dead wood stock is some 6-7 m3/ha (consistent with Table 1 values), so we expect current NFI data represent conditions where mortality in Romanian forest show important values. Fixed turnovers are applied for each biomass compartment (foliage, stemwood, branches and two belowground pools) by CBM runs. Text is added/improved on lines: a) explanation on how CBM deals with mortality in old stands is added under CBM description, line 150-on.

Comment 3.9: line 178 on Yasso model; the size of litter can be elaborated further, how it is applied in the study, particularly, comparing harvesting and extensive management scenario;

Response 3.9: Another correct observation. Text is added/improved on lines: Dead wood is split on two sub-pools by Yasso, we have assumed 100% of the mortality of standing merchantable stock is above 10 cm, so it is run by Yasso assuming as large deadwood pool dimensions.  

Comment 3.10: lines 209-210; if I understand correct that the same density is applied to all kind of dead wood, this is leading to significant overestimation of carbon stock in dead wood, because density of dead wood decreases with age and carbon stock in partly decomposed dead wood is significantly smaller. This should be clarified;

Response 3.10: Insightful question by the reviewer. CBM runs C stocks (not volume), so standing C stock and turnovers simulated implicitly take into consideration the actual wood density for each forest type (conversion of volume to biomass is done earlier in the simulation process, so there is no risk of introducing wrong C calculation). Moreover, in case of mixed forests a density is computed based on tree species participation. Finally, the inputs in both soil models is in C and not volume. An explanation is added on line 236-238.

Comment 3.11: lines 221-222; reference to Table 1 should be corrected;

Response 3.11: The reference was corrected.

Comment 3.12: chapter 2.4 generally; is it really so that non-woody litter is not considered in the calculation? It is also not clearly stated, how carbon input due to turnover of fine roots is considered;

Response 3.12: We agree that it was not clear and complete. Both models are litterfall input based, and the whole amount of litter generated annually by all biomass compartments is taken into consideration (indeed, not for bushes and other inferior taxa). Text is added/improved.  

Comment 3.13: line 277; how harvests are associated with increment? This statement should be changed to characterize proportion of harvested stands in comparison to stands / stock available for harvesting. Comparison of harvests and increment is useless information in projections of harvest rate;

Response 3.13: We agree that comparison of harvests and increment is useless information in projections turnovers and litterfall, so we also added info on the ratio of harvest from total standing stocks (as % of volume). The reason we selected is for the relevance from forestry perspective, i.e. sustainability, so we also keep the old information in the text. See also the response to the following issue raised for lines 280-282. Text is added/improved; the range of % of harvest in total standing stock volume is added in the text i.e., “(1) business as usual (BAU) scenario where the annual harvest was approximately 60% of the volume increment or between 0.1 and 0.14% of the standing stock in that year (ratios based on NFI’s estimates)”.

Comment 3.14: lines 280-282; The need for theoretical ranging of harvest projections is understandable, but how harvest projections are associated with increment? Maximum harvest projections should be associated with with availability of wood in mature stands and wood available for extraction in thinning. If there is no legal association between permitted harvest rate and increment, this scenario should be renamed to avoid misinterpretation of the assumptions, otherwise reference to the legal act should be added;

Response 3.14: Ratio of harvest to increment is the guiding principle in Romanian forestry, going beyond that result in unsustainable forest management on short run at least. It is strictly applied through planning the harvest at the most disaggregated forest administrative entity (majority of small forest owners are grouped in forest districts where there is a planning in place observing such guidance). Moreover, we do not only project harvest but rather various intensity of management (as from NF2 to NFI1 changes) to reach certain level of harvest. Although, as you mentioned we “hide” a very complex approach behind single term harvest. Text was added/improved on section 2.6; improved the understanding of the relation of increment to harvest.  

Comment 3.15: lines 325-326; it is not clearly stated, how stand types effect is considered in calculation and how it is avoided in climate stratification scenario. The term "sligtly" is too optimistic here, there is significant and systematic underestimation;

Response 3.15: The models’ results from each simulation are the same, they are aggregated in different way, either on only climate or only forest types, or on both. Text is added/improved on lines 368 to 372.

Comment 3.16: line 361; reference should be updated;

Response 3.16: The reference was updated.

Comment 3.17: Table 3; carbon stock in dead wood seems to be very small. This corresponds to 3-4 mha-1. Probably this is the reason for underestimation of carbon stock in the modelling exercise? If possible this value should be verified by other studies;

Response 3.17: Table 3. According to the background calculations the C stock estimated correspond to some 6-7 m3/ha, as most of the dead wood is in resinous-based forests (less accessible areas), much less in broadleaved forests (more accessible landscape). We provide the range for 95% confidence interval, so few negligible corrections occurred in the values, for all parameters presented in Table 3.

Comment 3.18: line 388; reference should be updated;

Done, it is a hyperlink issue with all tables and figures when MS was uploaded.

line 476; reference should be updated;

Done.

lines 526-527; it is also reasonable to mention in discussion that impact of non-woody litter and turnover of fine-roots may be underestimated in the calculation, particularly in scenario considering reduction of average leaf area index of woody vegetation due to harvests;

Response 3.18: The text was improved.

Comment 3.19: lines 530-531; such conclusions should be based on long-term observations / projections. Is it so that forests in central and southern Sweden doesn't contain carbon in soil after 800 years of clear-felling history? Both sides of the story should be reflected in the discussion;

Response 3.19: The text was improved.

Comment 3.20: lines 649-651; this conclusion is not substantiated by the study due to short period of projections. At least 2 generations of trees should be calculated to see real long term effect. The formulation "long term" in conclusions should be changed to particular calculation period.

Response 3.20: We understand the reviewer concern, so more cautious and realistic text was added.